# Pseudoislet Aggregation of Pancreatic β-Cells Improves Glucose Stimulated Insulin Secretion by Altering Glucose Metabolism and Increasing ATP Production

**DOI:** 10.3390/cells11152330

**Published:** 2022-07-29

**Authors:** Deborah Cornell, Satomi Miwa, Merilin Georgiou, Scott James Anderson, Minna Honkanen-Scott, James A. M. Shaw, Catherine Arden

**Affiliations:** 1Biosciences Institute, Newcastle University, Newcastle Upon Tyne NE2 4HH, UK; deborahcornell.11@gmail.com (D.C.); satomi.miwa@newcastle.ac.uk (S.M.); merilin.georgiou@newcastle.ac.uk (M.G.); 2Translational and Clinical Research Institute, Newcastle University, Newcastle Upon Tyne NE2 4HH, UK; scott_anderson2007@hotmail.co.uk (S.J.A.); minna.honkanen-scott@newcastle.ac.uk (M.H.-S.); jim.shaw@newcastle.ac.uk (J.A.M.S.)

**Keywords:** pancreatic β-cells, islets, pseudoislets, metabolism, connexin 36, glycolysis, oxidative phosphorylation

## Abstract

Appropriate glucose-stimulated insulin secretion (GSIS) by pancreatic β-cells is an essential component of blood glucose homeostasis. Configuration of β-cells as 3D pseudoislets (PI) improves the GSIS response compared to 2D monolayer (ML) culture. The aim of this study was to determine the underlying mechanisms. MIN6 β-cells were grown as ML or PI for 5 days. Human islets were isolated from patients without diabetes. Function was assessed by GSIS and metabolic capacity using the Seahorse bioanalyser. Connexin 36 was downregulated using inducible shRNA. Culturing MIN6 as PI improved GSIS. MIN6 PI showed higher glucose-stimulated oxygen consumption (OCR) and extracellular acidification (ECAR) rates. Further analysis showed the higher ECAR was, at least in part, a consequence of increased glycolysis. Intact human islets also showed glucose-stimulated increases in both OCR and ECAR rates, although the latter was smaller in magnitude compared to MIN6 PI. The higher rates of glucose-stimulated ATP production in MIN6 PI were consistent with increased enzyme activity of key glycolytic and TCA cycle enzymes. There was no impact of connexin 36 knockdown on GSIS or ATP production. Configuration of β-cells as PI improves GSIS by increasing the metabolic capacity of the cells, allowing higher ATP production in response to glucose.

## 1. Introduction

Insulin release by pancreatic β-cells in response to elevated blood glucose is an essential component of blood glucose homeostasis [1]. Diabetes develops when β-cells fail to release sufficient insulin to control blood glucose levels. In Type 1 diabetes, this is a result of auto-immune destruction of the β-cells, whilst in type 2 diabetes, loss of both β-cell function and mass in the face of peripheral insulin resistance leads to loss of blood glucose control [1,2]. Therapies to replace, regenerate or improve functional pancreatic β-cells mass are a major focus for the treatment of diabetes.

Pancreatic β-cells localise to the islets of Langerhans within the pancreas, in close association with other endocrine cells including glucagon-producing α-cells, somatostatin-producing δ-cells as well as ε- and Pancreatic Polypeptide (PP) cells [3]. The islet architecture is species-dependent with rodent islets consisting of a core of β-cells surrounded by a mantle of α- and δ-cells whilst human islets have a more heterogeneous distribution throughout the islet [3]. Despite these differences in islet architecture, both rodent and human islets display robust insulin secretion response when exposed to nutrient stimuli, particularly glucose [1]. Glucose stimulated insulin secretion (GSIS) is highly dependent on cell–cell communication between adjoining β-cells, with dispersed β-cells showing decreased GSIS compared to β-cells contained within intact rodent islets, a phenomenon reversed by re-aggregation of the β-cells [4,5,6]. Similarly, culture of mouse (MIN6) and human (EndoC-βH1) β-cell lines as ‘islet-like’ structures termed ‘pseudoislets’ (PI) improves the GSIS response when compared to their usual monolayer (ML) culture conformation [7,8,9,10]. It has been assumed the improved functionality of these 3D structures is due to co-ordination of individual β-cells through E-cadherin and gap junctions promoting connectivity between neighbouring cells, allowing synchronisation of intracellular Ca^2+^ oscillations between cells, evoking a larger GSIS response compared to cells with limited cell connections [11,12].

The responsiveness of β-cells to changes in physiological glucose concentrations is determined by a number of key β-cell characteristics which involve the expression of allowed vs. disallowed genes [13]. This includes high expression of glucokinase, pyruvate carboxylase, pyruvate dehydrogenase, glycerol 3-phosphate and the malate/aspartate shuttles in conjunction with low/negligible expression of low K_m_ hexokinases, lactate dehydrogenase and monocarboxylate transporter [13]. The specialised phenotype ensures sensing of changes around physiological glucose concentrations and also ensures maximal ATP production from glucose through tight coupling between glycolysis and mitochondrial glucose oxidation. The preferential shuttling of pyruvate into the mitochondria where it is further metabolised by the TCA cycle produces reducing equivalents in the form of NADH and FADH_2,_ which are used by the electron transport chain to fuel ATP synthesis [13,14]. This is essential to drive closure of the ATP-sensitive K^+^ channels which leads to membrane depolarisation and subsequent activation of voltage-dependent Ca^2+^ channels and downstream insulin exocytosis [13,14]. Inhibition of mitochondrial metabolism or uncoupling from ATP production, impairs GSIS [14,15], enforcing the critical role for mitochondrial metabolism in this pathway. Whilst these characteristics are evident in primary β-cells of both human and rodent origin, their maintenance in clonal β-cell lines is variable and has previously been shown to correlate with a decreased GSIS response [16,17]. Furthermore, the loss of GSIS response induced by chronic exposure to glucose and/or lipid, conditions mimicking type 2 diabetes, has been shown to be secondary to loss of mitochondrial function, decreased ATP production and increased oxidative stress [18,19]. Whether configuration of β-cells as 3D islets impacts on the metabolic characteristics and glycolytic-mitochondrial coupling of the β-cells is unclear. A previous study showed changes in protein expression of numerous metabolic enzymes involved in glycolysis, the TCA cycle and oxidative phosphorylation when MIN6 cells were configured as PI, although the impact of this configuration on metabolic flux was not fully investigated [20]. In this study, we sought to explore a role for cell–cell interactions in maintaining the characteristic features of β-cell metabolism and to determine the impact on metabolism flux.

## 2. Materials and Methods

### 2.1. Cell Culture

The MIN6 cell line were obtained from Dr. Jun-ichi Miyazaki (Kumamoto University Medical School, Japan). MIN6 cells (passage 22–26 (Figure 1B), passage 26–30 (all other figures)) were cultured as described previously [21]. ML and PI cultures were prepared by either culturing MIN6 on standard cell culture plates (for ML) or culturing in un-coated Petri dishes (for PI) for 5 days at a seeding density of 2.8 × 10^5^ cells/cm^2^. For hypoxic controls, MIN6 ML and PI were cultured for the final 24h of culture at 1% O_2_, 5% CO_2_ using a Sanyo MCO-19M incubator.

### 2.2. shRNA Knockdown of Connexin36

Lentiviral vectors expressing inducible shRNA against mouse connexin 36 were obtained from Dharmacon/Horizon Discovery (Cambridge, UK) (SMARTvector Inducible Lentiviral shRNA system using a mCMV promoter and turboGFP tag), with scrambled shRNA used as control. MIN6 cells were transfected with lentiviral particles expressing scrambled shRNA (Dharmacon, VSC6570) or shRNA against connexin 36 (Dharmacon, V3SM7672-235462855) at 2 × 10^6^ for 8 h prior to selection with 5 μg/mL puromycin for 10–14 days to generate stable cell lines. Expression of shRNA was induced in PI structures on day 2 of culture by the addition of 1 μg/mL doxycycline for 3 days.

### 2.3. Human Islet Isolation and Culture

Human islets were isolated from 3 non-diabetic donors in either the Human Islet Isolation Unit, King’s College, London, UK or the Newcastle University Transplant Regenerative Medicine Laboratory, with appropriate ethical approval. Research was performed with written donor-relative consent in accordance with the Declaration of Helsinki, and the protocol was approved by the Ethics Committee of MREC for Wales (05/MRE09/48). The clinical data associated with each islet preparation are summarised in Appendix A. Islets were maintained in CMRL media supplemented with 0.5% human albumin serum and 50 U/mL penicillin and 50 μg/mL streptomycin for 24 h prior to analysis [22].

### 2.4. Viability Staining

MIN6 ML or PI were incubated with 10 µg/mL propidium iodide and 10 µg/mL Hoechst-33342 for 10 min at room temperature. Following media change, cells were imaged using a Nikon TE2000 (Surbiton, UK) (× 20). Images were analysed using CellProfiler (www.cellprofiler.org) software to calculate percentage viability.

### 2.5. Glucose Stimulated Insulin Secretion Assay

MIN6 ML or PI were washed in Krebs-Hepes buffer (119 mM NaCl, 4.74 mM KCl, 2.54 mM CaCl_2_, 1.19 mM MgCl_2_, 1.19 mM KH_2_PO_4_, 25 mM NaHCO_3_, 10 mM Hepes (pH 7.4) and 0.5% bovine serum albumin (BSA)) [23]. Cells were incubated in Krebs-Hepes buffer at 0.5 mM glucose for 30 min at 37 °C then washed twice with Krebs-Hepes. Cells were then incubated for 1 h in Krebs-Hepes buffer containing either 5 mM or 25 mM glucose. The supernatant was removed and the insulin concentration measured using a high range rat insulin ELISA kit (Mercodia (Uppsala, Sweden)). The remaining cells were washed in PBS and extracted into 0.05% Triton-X 100 for protein quantification using the Bradford assay to normalise the insulin release/μg protein.

### 2.6. Enzyme Activity Assays

MIN6 ML or PI were extracted into extraction buffer (150 mM KCl, 3 mM Hepes, 1 mM dithiothreitol (DTT), 1 mM Benzamidine, 10 µL Protease Inhibitor Cocktail (Sigma Aldrich (Gillingham, UK))) and sonicated. Enzyme activity assays were performed using kinetic analysis for 10 min at 37 °C, absorbance 340 nm unless otherwise specified. Methods were modified from [24]:(i)Low K_m_ hexokinase (HK) and glucokinase (GK): main reagent (MR) = 50 mM Hepes (pH 7.8), 100 mM KCl, 2 mM MgCl_2_, 6 mM ATP/Mg^2+^, 1 mM NAD, 2 mM DTT, and 1.5 U/mL glucose 6-P dehydrogenase. A glucose concentration of 0.5 mM was used to measure low K_m_ HK activity and 50 mM glucose to measure total HK activity. GK activity was calculated by the difference in activity between these two concentrations.(ii)Phosphoglucoisomerase (PGI): MR = 50 mM Hepes, 1 mM MgCl_2_, 0.5 mM NAD, 2 mM fructose 6-phosphate, 1 mM DTT, and 2.5 U/mL glucose 6-P dehydrogenase.(iii)Phosphofructokinase 1 (PFK1): MR = 20 mM Tris, 100 mM KCl, 2 mM NH_4_Cl_2_, 3 mM MgCl_2_, 1 mM ATP/MgCl_2_, 0.16 mM NADH, 2 mM AMP, 10 mM fructose 6-phosphate, 1 mU/mL α-glycerophosphate dehydrogenase, 10 mU/mL triosephosphate isomerase, and 1 mU/mL aldolase.(iv)Glyceraldehyde 3-phosphate dehydrogenase (GAPDH): MR = 0.1 M triethanolamine (pH 8), 6 mM glycerate 3-phosphate, 0.18 mM EDTA, 1.12 mM ATP, 0.4 mM NADH, 5 mM MgSO_4_, and 14.8 U/mL Phosphoglycerokinase.(v)Aldolase (ALD): MR = 20 mM Tris, 100 mM KCl, 0.32 mM NADH, 10 mM fructose 1,6-bisphosphate, 1 mU/mL α-glycerophosphate dehydrogenase and 10 mU/mL triosephosphate isomerase.(vi)Pyruvate kinase (PK): MR = 0.1 M triethanolamine, 4 mM phospho(enol)pyruvic acid (PEP), 4 mM ADP, 0.4 mM NADH, 5 mM MgSO_4_, and 34 U/mL LDH.(vii)Lactate Dehydrogenase (LDH) activity was measured in main reagent containing 0.2 M KPi, 0.2 mM NADH, and 2.8 mM pyruvate.(viii)Pyruvate Carboxylase (PC): MR = 80 mM Tris, 2mM ATP, 16 mM Sodium pyruvate, 22 mM KHCO_3_, 9 mM MgSO_4_, 0.32 mM acetyl CoA, 0.16 mM NADH, and 10 U/mL malate dehydrogenase.(ix)Citrate Synthase (CS): MR = 50 mM Tris (pH 8), 0.3 mM Acetyl CoA, 0.24 mM oxaloacetate, and 0.2 mM DTNB. Absorbance was read at 412 nm.(x)Isocitrate Dehydrogenase (ICD): MR = 50 mM KPi (pH 7.4), 10 mM MgCl_2_, 2.5 mM isocitrate, and 0.25 mM NADP.(xi)α-ketoglutarate Dehydrogenase (αKG): MR = 100 mM Tris (pH 8), 0.5 mM NAD, 3 mM MgCl_2_, 0.2 mM Thiamine Pyrophosphate, 0.04 mM Coenzyme A, 2.5 µM rotenone, and 5 mM α-ketoglutarate.(xii)Malate dehydrogenase (MDH): MR = 0.2 M KPi, 0.2 mM NADH, and 2.78 mM oxaloacetate.

### 2.7. Seahorse Analysis

Cells were seeded 5 days prior to Seahorse analysis. ML were seeded at a density of 80,000 cells per well into a XFe24 cell culture microplate whilst PI were plated into Petri dishes at 2.8 × 10^5^ cells/cm^2^. One day prior to analysis, approximately 300 PI were placed in the centre of each well of a Seahorse islet capture plate. Cells were incubated in base media (Seahorse base media (Agilent (Stockport, UK)), supplemented with 1 mM glucose, 2 mM glutamate and 3% FBS) for 1h prior to loading onto the XFe24 analyser. Compounds were injected at the time point stated for final concentrations of 1 or 25 mM glucose, 10 mM pyruvate, 100 μM UK5099, 2 µM (ML) or 20 µM (PI) oligomycin, 0.5 μM FCCP, 5 μM rotenone, 0.5 µM antimycin A [25]. Readings were taken every 7 min. The duration between injections varied between ML and PI experiments due to preliminary studies indicating that PI took longer to reach a steady state, presumably due to their 3D morphology. Cell protein was extracted in 0.05% Triton-X 100 for quantification using the Bradford assay and used to normalise Seahorse readings. For calculations: ATP from oxidative phosphorylation was calculated using the OCR_ATP_: OCR_basal_-OCR_oligo_ where OCR_basal_ is the OCR measured after the respiration of the MIN6 has reached equilibrium after addition of glucose and OCR_oligo_ is the minimum OCR after addition of oligomycin [24]. ATP from glycolysis was based on ECAR_ATP_ where the ECAR due to CO_2_ production by mitochondrial metabolism has been subtracted using the equation described by [24]: ECAR_ATP_ = ECAR_tot_/BP-(10 (^pH-pK1^)/(1 + 10(^pH-pK1^)))(max H^+^/O_2_)(OCR_tot_-OCR_AA_) where ECAR_tot_/BP is the total cellular ECAR divided by the buffering capacity of the media and OCR_tot_-OCR_AA_ is the remaining OCR after mitochondrial metabolism is blocked by the addition of the complex III inhibitor, antimycin A [26,27]. ATP production is expressed as pmol ATP/min/mg protein.

### 2.8. Western Blotting

For assessment of hypoxia, MIN6 ML or PI were cultured for 24 h with 10 µM pimonidazole HCl (PIM) prior to protein extraction. Proteins were fractionated using 4–12% SDS/PAGE gels (Bio-Rad, Hertfordshire, UK) and transferred to PVDF. Membranes were probed with primary antibody (PIM: HP2-1000Kit, Hypoxyprobe (Burlington, MA, USA); GLUT2: NBP2-22218, Novus Biologicals (Cambridge, UK); HK I: SC-6517, Santa Cruz (Heidelberg, Germany); β-actin: 66009-1-Ig, Proteintech (Manchester, UK)) at 4 °C overnight. After incubation with secondary antibody conjugated to horseradish peroxidase, bands were detected using enhanced chemiluminescence. Immunoblots were quantified using ImageJ (imagej.nih.gov/ij).

### 2.9. RT-PCR

RNA was extracted using the High Pure RNA isolation kit (Roche Diagnostics (Welwyn Garden City, UK)) and cDNA was synthesized from 1 mg RNA with Maloney murine leukemia virus (Promega (Chilworth, UK)). Real-time RT-PCR was performed as in [28] using custom-designed taqman probes (ThermoFisher Scientific (Loughborough, UK), Appendix A). Relative mRNA levels were calculated by delta cycle threshold, corrected for the housekeeping gene (ribosomal protein lateral stalk subunit P0 (RPLP0)), and expressed relative to control.

### 2.10. Statistical Analysis

All statistical analysis was carried out on GraphPad Prism 6 software (San Diego, CA, USA). The mean and standard error mean were calculated and means were compared using an unpaired *t* test or one-way or two-way ANOVA followed by Bonferroni post hoc test as stated in the figure legends, to test for significance at *p* < 0.05.

## 3. Results

### 3.1. Configuration of MIN6 as Pseudoislets Improves Glucose-Responsiveness

MIN6 cells cultured on standard cell culture plastics formed as 2D ML whilst culture on un-treated plastic for 5 days resulted in the formation of 3D PI (Figure 1A). Configuration of young passage MIN6 (p22–p26) as PI did not improve the GSIS response compared to ML (Figure 1B). However, when replicated in MIN6 from slightly higher passages (p26–p30), the GSIS response was much greater in PI compared to ML (Figure 1C), which was quantified as a to 5.2 ± 1.1-fold stimulation in PI vs. 1.5 ± 0.3-fold in ML (Figure 1D). This was associated with a trend towards decreased basal insulin secretion (0.59 ± 0.1 μg/mg protein in ML; 0.44 ± 0.04 μg/mg protein in PI) (Figure 1C), although this did not reach significance. All subsequent experiments were performed on MIN6 passage 26–30 [29].

### 3.2. Pseudoislet Formation Increases ATP Production in Response to Glucose

Since appropriate GSIS is reliant on ATP production from glucose metabolism [13] we next explored whether 3D configuration of β-cells alters their metabolic profile and improves their capability to generate ATP in response to glucose stimulation. To test this, MIN6 were configured as ML and PI, and metabolic flux measured using the Seahorse bioanalyser. First, we assessed the effect of glucose on the oxygen consumption rate (OCR), which represents flux through mitochondrial oxidative phosphorylation [25]. Glucose caused a small increase in the OCR of ML (Figure 2A) but a more pronounced increase in PI (Figure 2B). Treatment with oligomycin decreased the OCR in both ML and PI (Figure 2A,B), consistent with the inhibition of ATP synthase. Using these data to calculate mitochondrial ATP production [23], it was apparent that glucose did not significantly increase mitochondrial ATP production in ML but caused a clear stimulation in PI (Figure 2C). This was associated with lower basal ATP production in PI, although this was not statistically significant.

We next assessed the impact of PI and ML culture on glycolysis by measurement of the extracellular acidification rate (ECAR), with appropriate calculation considering the ECAR contribution by mitochondrial CO_2_ production [30]. Glucose stimulation increased ECAR in both ML and PI (Figure 2D,E) but the response was much greater in PI. Although ECAR is often assumed to reflect anaerobic glycolysis through the generation of lactate, in cell types with high rates of mitochondrial metabolism there can be significant contribution to acid production from alternative pathways of H^+^ formation, namely pyruvate to bicarbonate formation in the mitochondria and via CO_2_ produced by the TCA cycle. We therefore used a previously validated calculation to correct the measured ECAR for respiratory acid production [25,26]. Using this methodology, ATP production from glycolysis showed much higher rates of glucose-stimulated ATP production in PI compared to ML (Figure 2F).

Combining the ATP production from OCR and glycolysis, PI showed improved glucose responsiveness compared to ML (Figure 2G), as indicated by the 5.1 ± 1.0-fold stimulation by glucose in PI vs. 1.6 ± 0.3-fold in ML (Figure 2H).

### 3.3. Interpreting the ECAR Rate in MIN6 ML and PI

The high ECAR rates and increased ATP production originating from glycolysis in our MIN6 models are inconsistent with previous literature emphasising low rates of pyruvate to lactate conversion in pancreatic β-cells and the importance of glycolytic-mitochondrial coupling [14,15,16]. We therefore further investigated the source of the ECAR signal in the MIN6 models.

First, we explored the impact of inhibiting mitochondrial oxidative phosphorylation using oligomycin. In most cell types, oligomycin stimulates ECAR as the cell switches to glycolysis in an attempt to maintain ATP production [31]. However, in MIN6 ML, oligomycin caused a clear decrease in ECAR (Figure 2D) whilst in PI, although there was an initial small increase in ECAR in response to oligomycin, this was followed by a gradual decrease (Figure 2E). These data are consistent with a large contribution of mitochondrial CO_2_ to the ECAR readout which is consistent with the high basal rates of OCR in these models (Figure 2A,B).

Next, we altered the initial stimulus from glucose to pyruvate and determined whether pyruvate stimulation could mimic the effect of glucose. Stimulation of ML with extracellular pyruvate increased OCR in ML to a similar rate to that of glucose (Figure 2A) but increased ECAR at a much smaller magnitude (Figure 2D). This suggests that the glucose-induced increase ECAR is not solely due to acid production via mitochondrial metabolism and that other pathways of acid production must be involved. It is important to note that the conversion of extracellular pyruvate to lactate is restricted due to the rapid depletion of cytosolic NADH in the absence of glycolysis, with the majority of the extracellular pyruvate being oxidised in the mitochondria [26]. As extracellular pyruvate had little effect on the OCR in PI (Figure 2C), it was not possible to interpret the lack of impact of pyruvate on ECAR.

Finally, we explored the potential fate of pyruvate in the MIN6 models, using an inhibitor of mitochondrial pyruvate uptake (UK5099) [32]. Treatment of MIN6 with UK5099 decreased OCR in both ML (Figure 3A) and PI (Figure 3B), consistent with inhibition of mitochondrial pyruvate uptake. Interestingly, UK5099 caused a large increase in ECAR in both ML (Figure 3C) and PI (Figure 3D). This suggests that in the absence of UK5099, a large proportion of pyruvate is being shuttled into the mitochondria for further metabolism as is expected in β-cells. However, the strong stimulation of ECAR in the presence of UK5099, suggest that both ML and PI have the capacity for significant lactate production via anaerobic glycolysis when mitochondrial pyruvate uptake is blocked.

Combining these findings, culture of MIN6 as PI increases their glucose responsiveness by stimulating ATP production, which is a consequence of both increased mitochondrial metabolism and also increased glycolysis.

### 3.4. Human Islets Show Increased OCR and ECAR in Response to Glucose Stimulation

Whilst the MIN6 PI model is considered a good representation of pancreatic islets, its major limitation is the reliance on a clonal β-cells which differ from endogenous β-cells. We therefore determined the contribution of OCR and ECAR to the glucose responsiveness of isolated human islets. Glucose stimulation of human islets increased OCR (Figure 4A) and also markedly stimulated ECAR (Figure 4C). There was no effect of pyruvate on either parameter. Oligomycin treatment decreased both OCR and ECAR (Figure 4A,B), suggesting that the ECAR readout may represent both anaerobic glycolysis and changes in oxidative phosphorylation, given the high basal OCR. Calculation of ATP production from glycolytic vs. mitochondrial sources show that glucose induced a greater increase in ATP production from glycolysis (Figure 4D) vs. mitochondrial metabolism (Figure 4B), with no response to pyruvate. Combining the ATP production from OCR and glycolysis, human islets showed good glucose responsiveness (Figure 4E,F). Although this is similar to the MIN6 PI model, it should be noted that glucose-stimulated ECAR was lower in magnitude in human islets compared to MIN6 PI (7.0 ± 1.9-fold vs. 33.8 ± 5.9-fold) despite similar increases in the OCR (2.1 ± 0.3-fold vs. 2.2 ± 0.3-fold).

### 3.5. Pseudoislet Formation Alters Expression and Activity of Glycolytic and TCA Cycle Enzymes

We next explored the underlying cause for the changes in metabolic profile of the PI vs. ML structures. When compared to ML, PI showed higher activity of several enzymes involved in glycolysis including phosphofructokinase 1 (PFK1), glyceraldehyde 3-phosphate dehydrogenase (GAPDH) and lactate dehydrogenase (LDH), whilst several other enzymes were unchanged (Figure 5A). Similarly, several enzymes involved in the TCA cycle also showed increased activity in PI structures vs. ML, including pyruvate carboxylase (PC), citrate synthase (CS) and α-ketoglutarate (αKG) (Figure 5B). These changes are consistent with the increased ATP production from both glycolytic and mitochondrial sources in PI.

Although not significant, the MIN6 PI showed a trend towards decreased basal rates of insulin secretion compared to ML (Figure 1C) and lower basal rates of OCR (Figure 2C). This was consistent with the lower activity levels of low K_m_ hexokinase activity (Figure 5A) and decreased protein expression of hexokinase I (Figure 5C). There was no significant change in glucokinase activity (Figure 5A). PI also showed higher expression of the glucose transporter GLUT2 (Figure 6D), consistent with improved glucose responsiveness.

### 3.6. Role for Hypoxia in Mediating Changes in Glucose Responsiveness

Previous studies on 3D islet-like structures have advised caution on long-term culture due to a developing hypoxic environment [33,34]. In the current study, we ensured that culture time was restricted to 5 days to minimise this effect, with PI maintaining good viability during this time period (89.2% ± 2.5% at day 5 (results not shown)). To assess hypoxia in the PI structures, we used pimonidazole, a bioreductive agent that forms protein adducts with thiol groups in hypoxic cells. Culture of ML and PI under hypoxic conditions for 24 h increased pimonidazole adduct formation (Figure 6A). There was no significant change in PIM adduct formation between ML and PI treated under normoxic conditions for 5 days (Figure 6A). However, real-time RT-PCR analysis of key hypoxic genes, showed that 5-day culture of PI caused a trend towards higher mRNA expression of key hypoxic genes, GLUT1, PDK1 and LDH, compared to ML (Figure 6B,C), suggestive of some degree of hypoxia in the 3D structures.

### 3.7. Connexin 36 Does Not Significantly Impact on GSIS or Metabolic Flux in Pseudoislets

Previous research has uncovered an important role for Connexin 36 (Cx36, also known as gap junction delta 2 protein (GJD2)), in maintaining Ca^2+^ synchronicity in 3D islet-like structures [12,35].

We therefore tested whether Cx36 was also important in maintaining the metabolic properties of the MIN6 PI. Cx36 was silenced in MIN6 PI structures for 2 days (from day 3 to day 5 of PI formation) by addition of doxycycline to MIN6 PI stably expressing shRNA against Cx36. Knockdown of Cx36 in this model was confirmed by Western blotting (Figure 7A) and had no impact on PI formation (results not shown). Silencing of Cx36 slightly increased basal insulin secretion, although this effect was not statistically significant (Figure 7B). There was no effect of Cx36 knockdown on the magnitude of the GSIS response (Figure 7B,C). Assessment of metabolic flux showed increased OCR in the absence of glucose stimulation following Cx36 knockdown (Figure 7D), but there was no impact of Cx36 knockdown on glucose-stimulated OCR or ATP production from mitochondrial metabolism (Figure 7E). There was also no significant effect of Cx36 knockdown on ECAR or glycolytic ATP production (Figure 7F,G). Combining ATP production from mitochondrial and glycolytic sources, there was no significant difference in either the basal or glucose-stimulated ATP production following Cx36 knockdown (Figure 7H,I), suggesting that Cx36 does not play a central role in the regulating the metabolic responses in PI structures.

## 4. Discussion

Previous studies have outlined the importance of both cell–cell communication [4,5,6,7,8,9,10,11,12] and the tight coupling of glycolytic to mitochondrial metabolism [13,14,15,16,17] to ensure maintenance of an effective GSIS response by pancreatic β-cells. In this study we used robust metabolic analysis to show that configuring pancreatic β-cells as 3D structures dramatically improves their capacity for ATP production in response to glucose stimulation, which is consistent with their increased GSIS response. Indeed, direct comparison of total ATP production with GSIS, shows that total ATP production was increased 5.1 ± 1.0-fold by glucose in PI but only 1.6 ± 0.3-fold in ML, which correlated tightly with a 5.2 ± 1.1-fold and 1.5 ± 0.3-fold increase in GSIS in PI and ML, respectively. These results suggest that changes in metabolic capability of PI is a major driver in the improvement in GSIS response.

Drilling down into the source of the increased ATP production in these 3D structures uncovered a complex phenotype. There was clear evidence of increased rates of glucose-dependent OCR and ECAR in MIN6 PI compared to ML. For most cell types, it is assumed that ECAR reflects changes in anaerobic glycolysis due to the conversion of pyruvate to lactate, which acidifies the media [30]. However, several studies show that ECAR can be influenced by other factors including the buffering capacity of the media, and the contribution of acid production from other pathways such as the TCA cycle through CO_2_ hydration to H_2_CO_3_, which in turn dissociates to HCO_3_^−^ + H^+^ [26,27,28,29,30,31]. We employed the corrective calculations described by Mookerjee et al. [26] to account for these factors and allow for calculation of ATP production from glycolysis and mitochondrial metabolism separately. Using this approach, it was obvious that whilst PI showed increased glucose-dependent ATP production from mitochondrial metabolism vs. ML (2.2 ± 0.3-fold vs. 1.2 ± 0.2-fold, respectively), there was a much greater impact of glucose on ATP production from glycolytic sources in PI vs. ML (33.8 ± 5.9-fold vs. 12.5 ± 3.8-fold, respectively). These data suggest that the improved GSIS response in PI is driven by increased ATP production from glycolysis, as well as increased mitochondrial metabolism. More detailed analysis of ECAR did indicate a potential influence of acid production from mitochondrial metabolism in the ECAR readout, since inhibition of oxidative phosphorylation by oligomycin failed to stimulate ECAR in our models. Instead, oligomycin decreased ECAR, which we believe is a consequence of a large reduction in OCR due to the high basal OCR rates. However, additional experiments indicate that CO_2_ production from mitochondrial metabolism is not sufficient to explain the glucose-induced increase in ECAR since: (i) there was a smaller increase in ECAR following pyruvate vs. glucose stimulation despite similar increases in OCR; (ii) inhibition of mitochondrial pyruvate uptake caused marked stimulation in ECAR. Given that the MIN6 models do express LDH [13] and thus are capable to converting pyruvate to lactate, we assume that the non-mitochondrial acid production partially reflects pyruvate to lactate conversion. It should also be noted that ECAR is also influenced by the pentose phosphate pathway (PPP), particularly in cells with high flux through this pathway. We did not measure contribution of the PPP to ECAR in this study and it remains feasible that the glucose-induced increased in ECAR, may also reflect changes in this pathway [26,30,31].

It is important to note that through conversion of ECAR and OCR into estimated rates of ATP production, we were able to assess the impact of glucose on total ATP production regardless of source. The finding that configuration of MIN6 as PI dramatically improves the capability of the β-cells to produce ATP in response to glucose suggests a previously unknown role for cell–cell interactions in mediating metabolic activity. Configuration as PI improves the capacity of the MIN6 cells to metabolise glucose and maximise ATP production at multiple regulatory steps. Although we did not measure glucose uptake, the increased expression of the GLUT2 glucose transporter is consistent with an improved capacity for glucose uptake and its downstream metabolism. PI also show alterations in their capacity for glucose phosphorylation. Pancreatic β-cells express glucokinase, also termed hexokinase IV, which differs from other hexokinase isoforms due to its low affinity for glucose and lack of product inhibition [13]. Glucokinase expression was not altered by PI formation. However, higher passage MIN6 cells also express low K_m_ hexokinases, which rapidly metabolise glucose at low concentrations (<1 mM). Configuration of MIN6 as PI decreased the activity of low K_m_ hexokinases and protein expression of hexokinase I. These findings are consistent with the trend towards lower basal insulin secretion and lower basal ATP production in PI, since less glucose will be metabolised at low glucose concentrations. PI also display higher activity of phosphofructokinase-1 (PFK-1), which catalyses the first enzymatic step in which glucose is committed to glycolysis and is consistent with higher rate of flux through glycolysis. The increase in LDH activity evident in PI is also consistent with the higher rates of glycolysis in these structures in response to glucose. The PI also show increased activity of several enzymes involved in the TCA cycle, with significant increases in both citrate synthase and α-ketoglutarate, in addition to trends towards increased activity of pyruvate carboxylase and malate dehydrogenase suggesting increased capacity for mitochondrial pyruvate metabolism. Although we did not assess the changes in the components of the electron transport chain, our findings of increased expression of glycolytic and TCA enzymes are consistent with increased ATP production via glucose metabolism. Importantly, these changes in enzyme expression are in agreement with a previous study showing increased protein expression of numerous glycolytic and TCA enzymes in PI structures versus ML using a proteomics approach [20]. The metabolic properties of PI are also altered in response to pyruvate, with ML showing uptake and metabolism of extracellular pyruvate whilst both MIN6 PI and human islets appearing unresponsive to pyruvate stimulation. This is consistent with previous reports of absence of MCT transporters in islets [36] and suggests that configuration of MIN6 as PI, perhaps reverts the cells back to towards a more typical β-cell phenotype. In our hands, configuration as PI improved the GSIS response only at passages >25, with PI from lower passage MIN6 showing little difference in GSIS response. Other studies have shown improvements in function at lower passages [20], although many groups do seem to utilise higher-passage MIN6 for PI studies [7,8].

The underlying mechanisms driving the changes in metabolic flux of PI appears complex. Due to their 3D structure, PI are more susceptible to hypoxia compared to their ML counterparts. Oxygen diffusion through the 3D islet structure is maintained by a dense capillary network but this will be disrupted under in vitro culture of islets and is absent during formation of MIN6 PI [33,34]. Continued static culture of MIN6 PI for >1 week alters oxygen and nutrient diffusion, leading to the development of necrotic cores [34]. In our study, we limited cell culture to a maximum of 5 days to minimise the development of hypoxia. Using these conditions, we saw no evidence for hypoxia in PI using pimonidazole staining, despite strong staining for our positive hypoxic controls. However, we did see an increase in expression/activity of several enzymes that make up the hypoxic gene signature [37], including GLUT1, PDK1 and LDH, indicating a potential hypoxic environment. Certainly, the increase in anaerobic glycolysis indicated by higher ECAR, is consistent with a hypoxic response. However, this mechanism is also usually associated with an increase in basal rates of insulin secretion [38]. In our model, there was no increase in the basal rate of either GSIS or glucose-stimulated ATP production in PI, but rather a trend towards decreased basal flux, which is inconsistent with these metabolic changes being secondary to hypoxia. It remains possible that the changes in metabolic flux induced by PI formation represent a mixed picture of a hypoxic response and more complex metabolic pathway signalling. We predicted that cell–cell interactions via Cx36 may be involved in the regulation of metabolic flux in PI. However, genetic knockdown of Cx36 did not reverse the metabolic properties of the PI structures. There was a trend towards an increase in basal insulin secretion which was accompanied by a small increase in basal ATP production from mitochondrial metabolism. However, the responsiveness of the PI to glucose stimulation remained similar following Cx36 knockdown, suggesting communication via this gap junction is not involved in the switch to improved metabolic function.

The MIN6 PI model has been utilised by many groups due to its improved GSIS responsiveness over ML culture and is often deemed to be a robust alternative for freshly isolated islets. In this study, we used techniques similar to those previously described to generate MIN6 PI [7,8] but restricted culture time to 5 days to limit any detrimental effects on cell viability and utilised state of the art methodologies and rigorous data analysis to assess the impact on metabolic flux. Our finding that a large component of glucose-stimulated ATP production in PI is derived from glycolysis, and less so mitochondrial glucose oxidation, was surprising, although consistent with previous proteomic analysis [20]. To determine whether this was restricted to the MIN6 model, we also measured OCR and ECAR rates in isolated human islets and calculated ATP production from glycolysis and mitochondrial metabolism using corrective calculations [26]. Human islets showed a large impact of glucose on ATP production from glycolysis (7.0 ± 1.9-fold) with a smaller impact on ATP production from mitochondrial metabolism (2.1 ± 0.3-fold), consistent with a high rate of anaerobic glycolysis in the 3D model. However, the contribution of glycolysis to ATP production in response to glucose stimulation was less prominent than in the MIN6 model. Furthermore, oligomycin decreased ECAR rates which is suggestive of significant contribution of mitochondrial metabolism to ECAR secondary to the high basal OCR of these models. These data suggest a component of glycolysis contributing to ATP production in human islets. Since these islets were kept in culture for 2–3 days prior to analysis, it is difficult to determine whether this fully recapitulates the human islet in vivo. It is possible that the phenotype of increased glycolysis in PI is a consequence of the fragility of the MIN6 model and additional work is required to determine if this mechanism is evident in primary cell models.

## 5. Conclusions

To conclude, our study uncovers an important role for cell–cell contacts in the regulation of metabolic flux in pancreatic β-cells and shows that configuration of MIN6 as PI leads to fundamental changes in the metabolic characteristics of the β-cells, which dramatically improves their capacity for ATP production in response to glucose. It appears that this change in ATP production is sufficient to explain the improvement in GSIS response in the MIN6 model. Further work is required to understand the underlying mechanisms, particularly since this does not appear to be entirely driven by hypoxia or Cx36, and to determine whether this mechanism is restricted to the MIN6 model.

## Figures and Tables

**Figure 1 cells-11-02330-f001:**
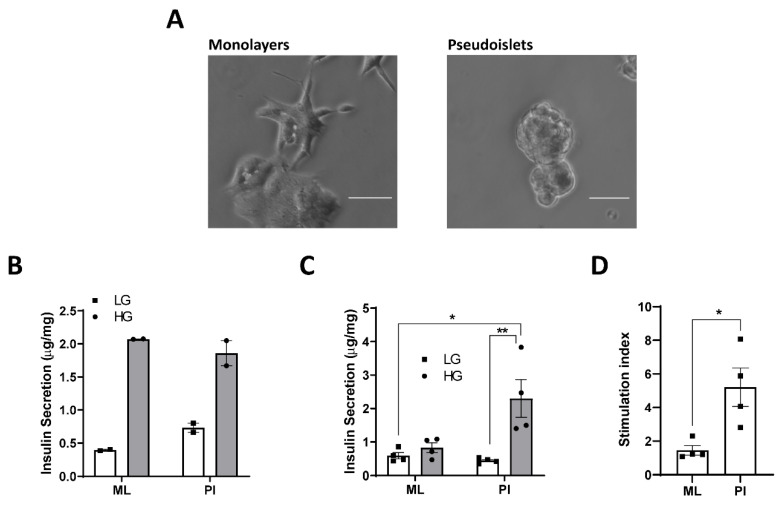
Configuration of MIN6 as pseudoislets improves GSIS. MIN6 cells (passage 22–26 (**B**) or 26–30 (**C**,**D**)) were cultured in either standard cell culture plates (Monolayers, ML) or un-coated Petri-dishes (Pseudoislets, PI) for 5 days. (**A**) Cells were imaged using light microscopy. (**B**,**C**). Insulin secretion at low (LG, 3–5 mM) or high (HG, 25 mM) glucose was determined in ML and PI for 1 h and expressed as μg insulin/h/mg protein. n = 2 (**B**) or 4 (**C**). (**D**) The stimulation index was calculated as the ratio of stimulated (HG) to basal insulin secretion (LG). n = 4. * *p* < 0.05, ** *p* < 0.01 using a 2-way ANOVA followed by Bonferroni post hoc test (**C**) or an unpaired student *t*-test (**D**). Scale bars represent 100 μm.

**Figure 2 cells-11-02330-f002:**
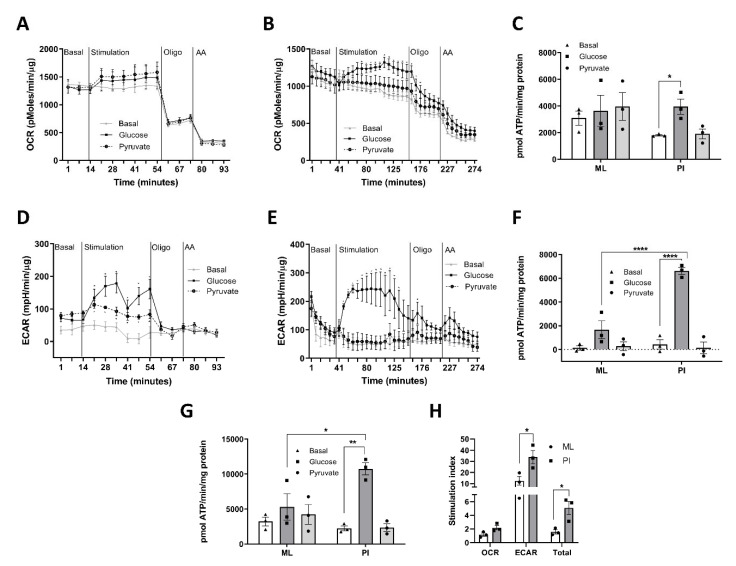
Pseudoislet formation increases ATP production in response to glucose. MIN6 cells (passage 26–29) were cultured in Seahorse 24-well plates for 5 days (ML) or un-coated Petri-dishes for 4 days before transfer to a 24-well islet capture plate for 24 h (PI). The Oxygen Consumption Rate (OCR) and Extracellular Acidification Rate (ECAR) were measured using the Seahorse XFe24 well analyser with additions of either media only (Basal), Glucose (25 mM), Pyruvate (10 mM) at time point 1 followed by Oligomycin (2 μM for ML, 20 μM for PI) at time point 2 or antimycin A (0.5 μM) at time point 3, as indicated. (**A**) OCR for ML expressed as pmol/min/μg protein. (**B**) OCR for PI expressed as pmol/min/μg protein. (**C**) ATP production from OCR as calculated outlined in the methods and expressed as pmol ATP/min/mg protein. (**D**) ECAR for ML expressed as mpH/min/μg protein. (**E**) ECAR for PI expressed as mpH/min/μg protein. (**F**) ATP production from ECAR expressed as pmol ATP/min/mg protein. (**G**) Total ATP production from OCR and ECAR expressed as pmol ATP/min/mg protein. (**H**) Stimulation index was calculated as the ratio of stimulated (glucose) to basal ATP production (basal). n = 3. * *p* < 0.05, ** *p* < 0.01, **** *p* < 0.001 using a 2-way ANOVA followed by Bonferroni post hoc test.

**Figure 3 cells-11-02330-f003:**
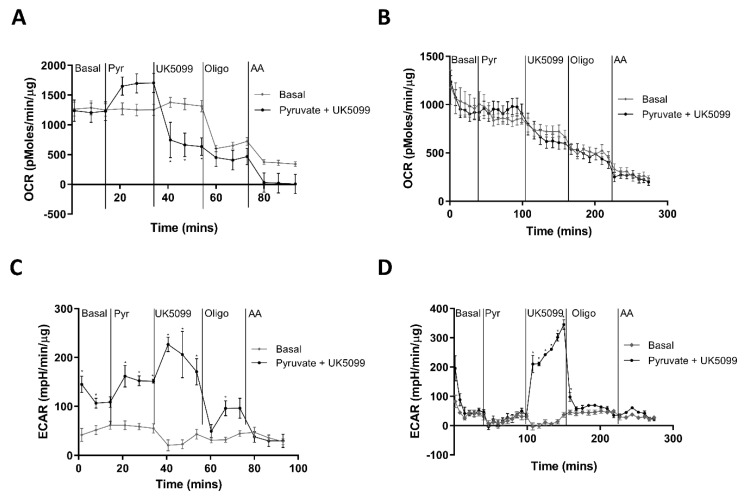
ECAR reflects changes in lactate production and mitochondrial metabolism. MIN6 cells (passage 26–29) were cultured in Seahorse 24-well plates for 5 days (ML) or un-coated Petri-dishes for 4 days before transfer to a 24-well islet capture plate for 24 h (PI). (**A**–**D**) The Oxygen Consumption Rate (OCR) and Extracellular Acidification Rate (ECAR) were measured using the Seahorse XFe24 well analyser with additions of either media only (Basal), Glucose (25 mM), Pyruvate (10 mM) at time point 1 followed by UK5099 (100 μM) at time point 2, oligomycin (2 μM for ML, 20 μM for PI) at time point 3 and antimycin A (0.5 μM) at time point 4, as indicated. (**A**) OCR for ML expressed as pmol/min/μg protein. (**B**) OCR for PI expressed as pmol/min/μg protein. (**C**) ECAR for ML expressed as mpH/min/μg protein. (**D**) ECAR for PI expressed as mpH/min/μg protein. n = 4. * *p* < 0.05 using a 2-way ANOVA followed by Bonferroni post hoc test.

**Figure 4 cells-11-02330-f004:**
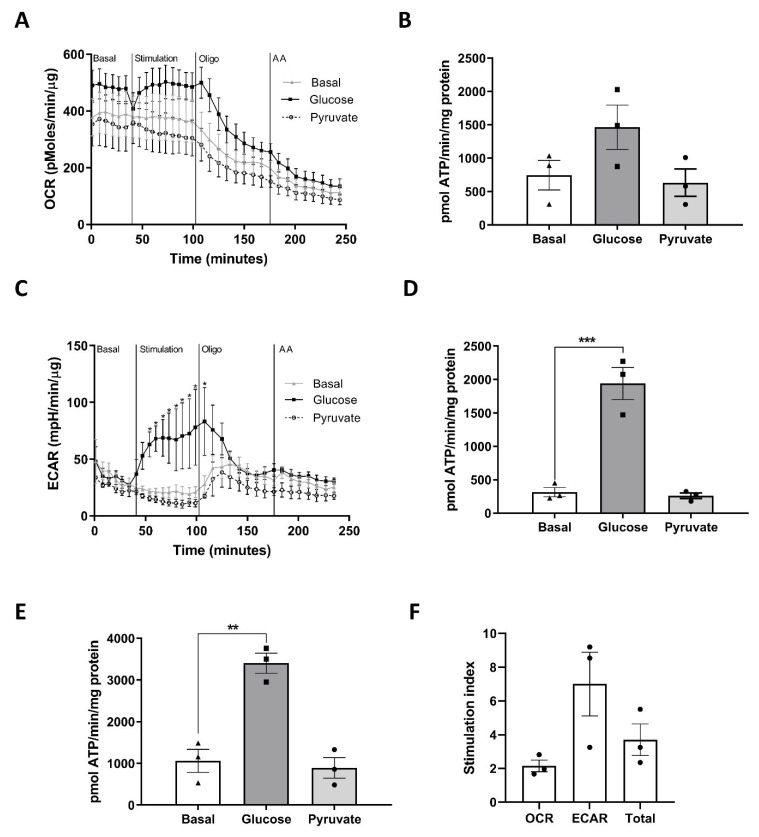
Human islets show increased OCR and ECAR in response to glucose stimulation. Human islets were cultured for 24 h following isolation prior to transfer into a 24-well islet capture plate for a further 24 h. ECAR and OCR was measured using the Seahorse XFe24 well analyser with additions of Glucose (25 mM), Pyruvate (1 mM), Oligomycin (20 μM) or antimycin A (0.5 μM) as indicated. (**A**) OCR is expressed as pmol/min/μg protein. (**B**) ATP production from OCR expressed as pmol ATP/min/mg protein. (**C**) ECAR is expressed as mpH/min/μg protein. (**D**). ATP production from ECAR expressed as pmol ATP/min/mg protein. (**E**). Total ATP production from OCR and ECAR. (**F**) Stimulation index was calculated as the ratio of stimulated (glucose) to basal ATP production (basal). n = 3. ** *p* < 0.01, *** *p* < 0.005 using a one-way ANOVA followed by Bonferroni post hoc test.

**Figure 5 cells-11-02330-f005:**
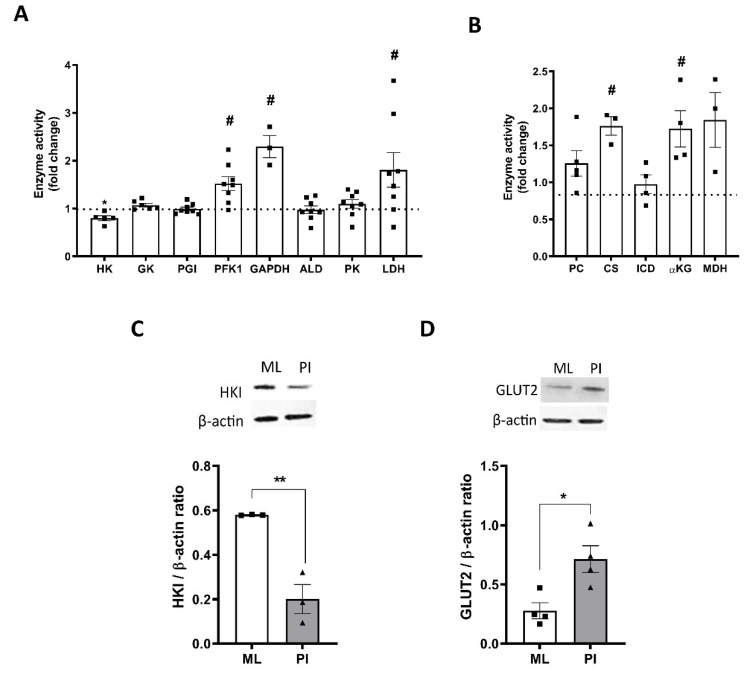
Pseudoislet formation alters expression and activity of glycolytic and TCA cycle enzymes. MIN6 cells (passage 26–29) were cultured in either standard cell culture plates (ML) or in un-coated Petri-dishes (PI) for 5 days. (**A**,**B**) Changes in enzyme activity for glycolytic enzymes (**A**) or TCA cycle enzymes (**B**) was determined on cell lysates from ML and PI and activity corrected for protein content. (**C**,**D**). Changes in protein expression of HK (**C**) and GLUT2 (**D**) were determined using Western blotting and expressed relative to β-actin. Results are expressed as fold change relative to ML (**A**,**B**) or as GLUT2 or HK: β-actin ratio (**C**,**D**). n = 3–8. * *p* < 0.05, ** *p* < 0.01, # *p* < 0.05 using an unpaired *t*-test.

**Figure 6 cells-11-02330-f006:**
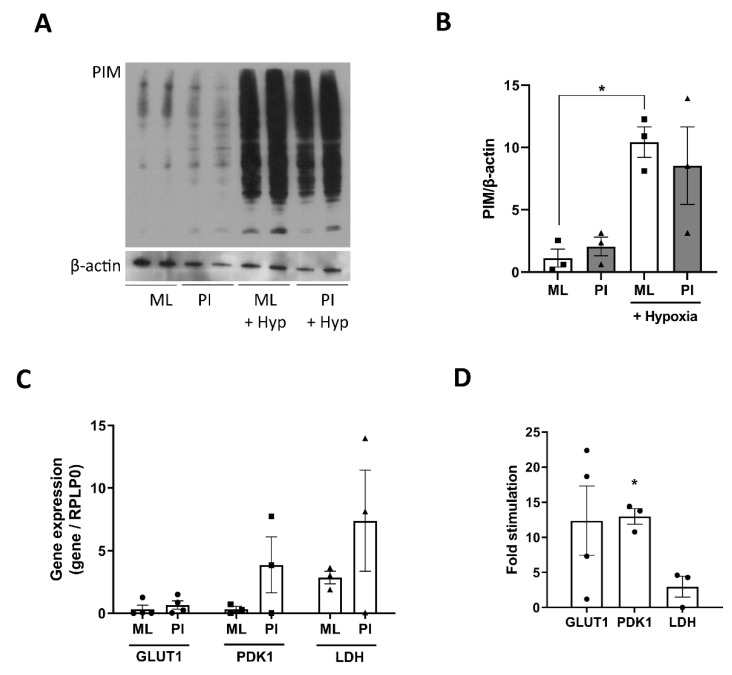
Pseudoislets show upregulation of some hypoxic markers. MIN6 cells (passage 26–30) were cultured in either standard cell culture plates (ML) or in un-coated Petri-dishes (PI) for 5 days. For assessment of hypoxia, ML and PI were cultured either under normoxia or hypoxic (1% O_2_) for the final 24 h of culture. (**A**) Hypoxia was assessed by Western blotting for PIM adjuncts and expressed relative to β-actin. (**B**,**C**) Changes in gene expression was determined using real-time RT-PCR and expression corrected for RPLP0. Results are expressed as fold change relative to ML (**B**) or as gene of interest: RPLP0 ratio (**C**), or as stimulation index (**D**). n = 3. * *p* < 0.05 using an unpaired *t*-test.

**Figure 7 cells-11-02330-f007:**
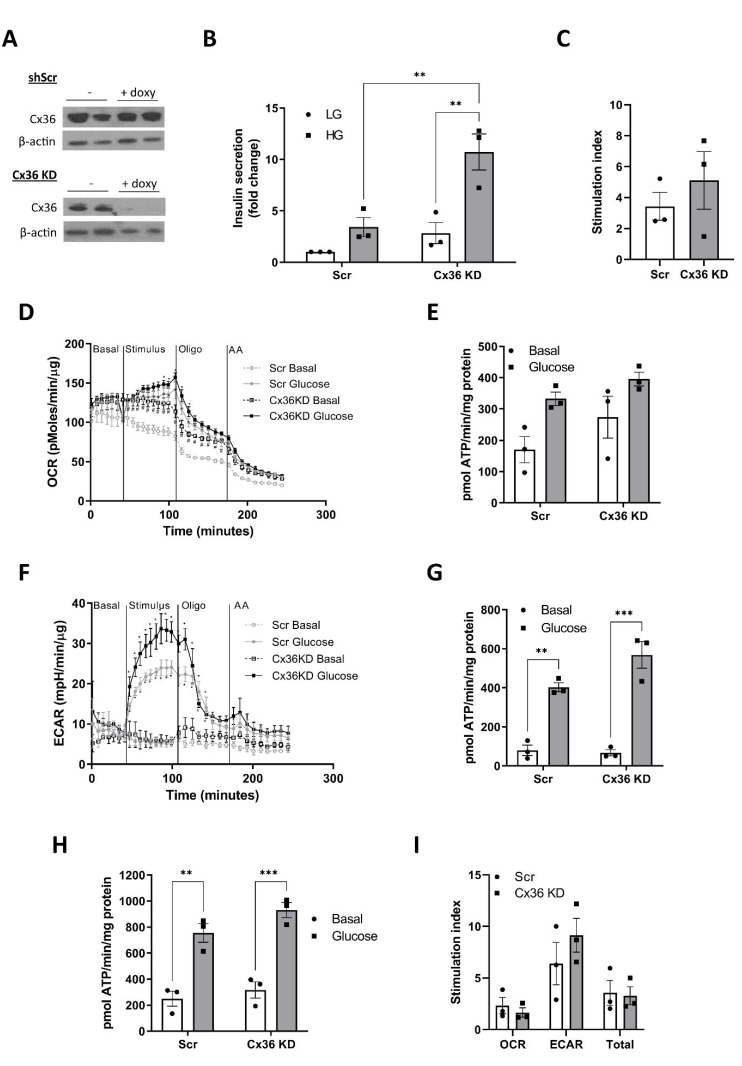
Connexin 36 does not significantly impact on GSIS or metabolic flux in pseudoislets. MIN6 cells expressing inducible scrambled shRNA (Scr) or shRNA against connexin 36 (Cx36 KD) were generated by transfection with lentiviral constructs followed by selection with 1 μg/mL puromycin. Cells were cultured in un-coated Petri dishes to generate PI for 3 days prior to the addition of 1 μg/mL doxycycline for a further 48 h to induce expression of shRNA constructs. (**A**) Protein expression of Cx36 was determined by Western blotting. Images are representative of 2 experiments (**B**) Insulin secretion at low (LG, 3–5 mM) or high (HG, 25 mM) glucose was determined in ML and PI for 1 h and expressed as fold change relative to Scr LG. (**C**) The stimulation index was calculated as the ratio of stimulated (HG) to basal insulin secretion (LG). (**D**–**I**) MIN6 PI were transferred to Seahorse islet capture plates for 24 h prior to measurement using the Seahorse XFe24 well analyser with additions of Glucose (25 mM), Oligomycin (20 μM) or antimycin A (0.5 μM) as indicated. (**D**) OCR expressed as pmol/min/μg protein. (**E**) ATP production from OCR expressed as pmol ATP/min/mg protein. (**F**) ECAR is expressed as mpH/min/μg protein. (**G**) ATP production from ECAR expressed as pmol ATP/min/mg protein. (**H**) Total ATP production from OCR and glycolysis expressed as pmol ATP/min/mg protein. (**I**) Stimulation index was calculated as the ratio of stimulated (glucose) to basal ATP production (basal). n = 3. ** *p* < 0.01, *** *p* < 0.005 using a 2-way ANOVA followed by Bonferroni post hoc test.

## Data Availability

The datasets generated during the current study are available from the corresponding author upon reasonable request.

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
