# Peer review of "Pseudoislet Aggregation of Pancreatic β-Cells Improves Glucose Stimulated Insulin Secretion by Altering Glucose Metabolism and Increasing ATP Production"

_cells, 2022, doi:10.3390/cells11152330_

Round 1

Reviewer 1 Report

By comparing cultured β-cells as 3D pseudoislets with their 2D monolayer form, the authors tried to explore a role of cell-cell connectivity in maintaining the characteristic features of β-cell metabolism and to determine the impact of cell-cell connections on metabolism flux.

The paper is written in accordance with all the standards expected in this scientific field. The experiment was realized through a series of modern techniques, with results that quantitatively and qualitatively exceed the usual practice. A detailed discussion of the obtained results is given, which logically connects current knowledge with the original contribution of this manuscript. With all the above in mind, I suggest that the manuscript be published without further changes.

Author Response

We thank the reviewer for the supportive comments.

Reviewer 2 Report

This study presented advantage of pseudo islet formation in response to glucose stimulation. However, it is not clear whether this report has differences and advantages compared with previous studies.

In addition, this study used human islets suggestive of confirmation of the Min6 study, which is so nice. However, conversely, if human islets will be dispersed to single cells culturing in monolayer, what will  happen is not presented. (Maybe so difficult study, but essentially required.)

Author Response

1. We have included a statement on the advantages that our study offers over previous studies (line 543-545). This is in addition to the several point we already include regarding the novelty of our study (including lines 443, 480, 513, 521, 565).

2. We had hoped to include a section of work where we explored the impact of dissociating human islets on the metabolic properties of human beta-cells. However, the process of dissociating human islets and measuring the metabolic properties of isolated beta-cells is complex. In our hands, the enzymatic disruption of islets decreased cell viability.  We therefore felt that it would be difficult to separate the impact of changes in cell conformation (2D vs 3D) from the invasive impact of enzymatic destruction on the metabolic function of the beta-cells. 

Reviewer 3 Report

Dear Authors, 

the work presented by you presents very interesting results on the mechanisms responsible for the functioning of pancreatic beta cells. The study is meticulously prepared and carried out. I recommend the above article for publication.

Author Response

(The authors gave the same response as above.)

Reviewer 4 Report

In the manuscript by Deborah Cornell et al, the authors compared the effect of 3D pseudoislets and 2D monolayer (ML) culture on glucose-stimulated insulin secretion (GSIS), and demonstrated the improved GSIS of 3D pseudoislets. Furthermore, a role for cell-cell connectivity in maintaining the characteristic features of β-cell metabolism and the impact of cell-cell connections on metabolism flux is investigated using the 3D pseudoislets model. It was shown that culturing MIN6 as PI improved GSIS; higher glucose-stimulated oxygen consumption and extracellular acidification rates in MIN6 PI partially due to increased glycolysis and no impact of connexin knockdown on GSIS. All the data demonstrate the application and advantages of MIN6 PI in the study of GSIS.

Major concern:

1.       The differences between cells cultured monolayer and 3D are not only the cell communications. It is inaccurate to draw a conclusion that this study uncovers the effect of cell-cell contacts on GSIS. The conclusion should be more accurate.

2.       It seems the monolayer cultured MIN6 lost the response to high glucose at passage 26-30 compared with passage 22-26. Did the author check the differences between the cells of passage 26-30 and 22-26? Are there morphology or other changes?

Minor concern:

1.       The legend for figure 2B is wrong.

2.       Line 252: redundant period.

3.       Is there any data of HK2 expression in the MIN6 PI compared with 2D monolayer?

Author Response

We thank the reviewer for their constructive comments and have amended our manuscript accordingly:

1. We have altered our conclusion as suggested by the reviewer. Instead of referring to changes in cell-cell communication, we instead refer to changes in cell-cell interactions or as configuration of beta-cells as monolayers vs pseudoislets (lines 82, 84, 437, 480, 534, 567).

2. We did not note any obvious morphological changes between passages 22-29. We have noted a change in morphology in MIN6 at passages >30, changing from rounded cells to more irregular elongated cells. We did not measure any other parameters other than insulin secretion in the current study. However, in past studies using MIN6 cells we have consistently seen a decrease in GSIS in MIN6 cells above passage 26, which is accompanied by an increase in basal insulin secretion. In some unpublished work, we showed that increasing passage number was associated with an increase in low-Km hexokinase activity (3-fold between passage 22 vs 26, 9-fold between passage 22 vs 29), whilst glucokinase activity remained largely unchanged. Consistent with this was an increase in glucose phosphorylation at basal (5mM) glucose concentrations (19-fold between passage 22 vs 29) (Arden et al, unpublished data). We believe that this is a consequence of the loss of the differentiated beta-cell phenotype as previously reported (Yamato et al, PMID: 23560115).

1. We have corrected the error in the legend for Figure 2B (line 251). Thank you for pointing this out.

2. We have removed the redundant period (line 255).

3. Unfortunately we did not measure HK2 mRNA expression in the study as it was not part of the usual hypoxic gene signature utilised by the team. We did attempt to assess HK2 at the protein level using western blotting. Unfortunately, we were unable to detect any bands using a HK2 specific antibody.